# Maize Leaf Disease Identification Based on YOLOv5n Algorithm Incorporating Attention Mechanism

**Li Ma** [1], **Qiwen Yu** [1], **Helong Yu** [1] **and Jian Zhang** [2,3,*]

1    College of Information Technology, Jilin Agricultural University, Changchun 130118, China
2    Faculty of Agronomy, Jilin Agricultural University, Changchun 130118, China
3    Department of Biology, University of Columbia Okanagan, Kelowna, BC V1V 1V7, Canada
*    Correspondence: jian.zhang@ubc.ca

**Abstract:** Maize diseases are reported to occur often, and are complicated and difficult to control, which seriously affects the yield and quality of maize. This paper proposes an improved YOLOv5n model incorporating a CA (Coordinate Attention) mechanism and STR (Swin Transformer) detection head, CTR_YOLOv5n, to identify common maize leaf spot, gray spot, and rust diseases in mobile applications. Based on the lightweight model YOLOv5n, the accuracy of the model is improved by adding a CA attention module, and the global information acquisition capability is enhanced by using TR2 as the detection head. The average recognition accuracy of the algorithm model can reach 95.2%, which is 2.8 percent higher than the original model, and the memory size is reduced to 5.1MB compared to 92.9MB of YOLOv5l, which is 94.5% smaller and meets the requirement of being light weight. Compared with SE, CBAM, and ECA, which are the mainstream attention mechanisms, the recognition effect we used is better and the accuracy is higher, achieving fast and accurate recognition of maize leaf diseases with fewer computational resources, providing new ideas and methods for real-time recognition of maize and other crop spots in mobile applications.

**Keywords:** deep learning; attention mechanism; maize leaf disease; digital agriculture





## 1. Introduction

Because it is one of the three main food crops and a significant source of revenue for many farmers cross the world, maize, which has a high nutritional value, continues to play a significant role in addressing the issue of human food supply today [1]. Data show that 60% of maize in China is used as feed for livestock and poultry industries, 30% is used for industrial purposes, such as chemical, pharmaceutical, and paper making, and the remaining 10% is used for direct consumption by people. Mazie occupies an important position in the agricultural production and economic development of China. It is evident that, together with rice and wheat, maize will be the food crop with the largest production demand in the future. Therefore, increasing maize production and maintaining high quality is important to China's agricultural industry. Among the many factors affecting maize production, the problem of maize pests and diseases has the greatest negative impact on its production and quality, and once maize pests and diseases occur, they can cause varying degrees of yield reduction and quality decline, seriously affecting the economic benefits of producers and the industry as a whole [2].

At present, the category identification of maize diseases in China is based on the empirical judgment of crop pathologist experts in the field and technicians specialized in plant protection; therefore, technicians need to have good observation skills and rich experience to accurately identify the category of diseases [3]. This traditional disease identification method, which relies on individual experience, has a large limitation. Additionally, when there are too many samples to test with many different disease types, subsequently, there is a higher chance of inaccuracy in the identification process due to human errors.

Recent years, with the high-speed development of big data analysis technology and GPUs (Graphics Processing Units), the computing power of computers has been improved, and deep learning techniques have been developed rapidly and have been used in many applications such as agricultural pests and diseases [4]. Yinglai Huang et al. [5] replaced $7 \times 7$ convolutional kernels in the first convolutional layer of the conventional ResNet-50 model with three $3 \times 3$ convolutional kernels; they used the LeakyReLU activation function instead of the ReLU activation function and changed the order of the batch normalization layer, activation function, and convolutional layer in the residual block. The improved network obtained a 98.3% correct rate in maize leaf disease image classification. Haoyu Wu [6] proposed to construct a two-channel convolutional neural network based on VGG and ResNet. By adjusting the parameters of the two-channel convolutional neural network, the accuracy of identifying maize leaf disease types in the validation set can reach 98.33%, while the VGG model can reach 93.33%. Chao Wang et al. [7] proposed a method based on ResNet (Residual Neural Network) deep learning network for maize disease recognition, using ResNet as the main model for maize disease recognition, and found that the highest classification accuracy of 92.82% was obtained with ResNet50 at a batch size of 32 and epoch number of 16.

Azlah, M.A.F. et al. [8] mainly reviewed the advantages of each classifier and compared their compatibility with different leaf features recognition process. Koklu, M. et al. [9] conducted a deep learning-based classification by using images of grapevine leaves. The most successful method was obtained by extracting features from the Logits layer and reducing the feature with the chi-squares method. The most successful SVM kernel was Cubic. The classification success of the system has been determined as 97.60%. Argüeso, D. et al. [10] introduced Few-Shot Learning (FSL) algorithms for plant leaf classification using deep learning with small datasets. The FSL method outperformed the classical fine-tuning transfer learning, which had accuracies of 18.0 (16.0–24.0)% and 72.0 (68.0–77.3)% for 1 and 80 images per class, respectively.

Although there are many recognition techniques based on deep learning technology and all of them work well, there are some problems among them, such as less small-scale target data, larger memory consumption of the model, and being unsuitable for mobile deployment.

This paper, therefore, investigates the problem of disease in maize leaves, applying the current deep learning technology to design an experimental study in the hope that farmers will be able to rely on their mobile phones in the field to identify diseases on maize in a timely and effective manner, thus alleviating the problems of reduced yields and reduced quality of maize. As the ultimate goal of our research is to help farmers to identify maize diseases in real-time in the field with a mobile device on their person, the light weight and high accuracy of the model is the focus of this paper.

Currently, the commonly used target detection networks include Faster R-CNN [11], SSD [12], YOLO series [13–16], etc. Among them, the YOLO network model belongs to a one-stage target detection algorithm with a simple structure, small computation, and fast operation speed, which is widely used in crop disease identification research. Among them, YOLOv5n is the latest model of the YOLOv5 series network, which has the advantages of high detection accuracy, fast inference speed, and small storage space, and is suitable for deployment in mobile for real-time detection. In this paper, we propose a regional detection model for maize leaf diseases based on YOLOv5n: CTR_YOLOv5n, which accelerates the model convergence speed, improves the model generalization ability, and enhances the recognition accuracy and detection speed of the model, taking three common maize leaf diseases, blotch disease, gray spot, and rust, as the research objects.

## 2. Materials and Methods

### 2.1. Construction of the Data Set

The target dataset used for the experiments in this paper is maize leaf diseases. Among the various maize leaf diseases, the most common and representative ones are blotch

disease, gray spot, and rust, and images of the three leaf diseases, as well as images of healthy maize leaves, are used as the dataset.

(a)    Maize blotch disease

The disease produces spots on the leaves, such as long rhombus shape. The color is generally brown or yellow-brown, and the rhombus-shaped spots are generally 5–10 cm long and 1 cm wide, approximately. The disease will gradually expand when the disease is serious, and even lead to leaf death.

(b)    Maize rust

The disease mainly occurs on maize leaves, on both sides of the leaf, and causes scattered or aggregated growth of round, yellow-brown, powdery spots and scattered rust-colored powder, that is, the summer spore mounds and summer spores of the pathogenic bacteria. Later in the season, round, black winter spore mounds and winter spores grow on the spots.

(c)    Maize gray spot disease

The disease initially forms oval to rectangular gray to light brown spots on the leaf surface without obvious margins, turning brown later. The spots are mostly confined between parallel leaf veins and are 4 to 20 × 2 to 5 (mm) in size. When the humidity is high, the abaxial surface of the spot produces gray moldy material, that is, the conidiophore and conidia of the disease.

(d)    Healthy maize leaves

Leaf blade flattened and broad, leaf sheath with transverse veins; ligule membranous, about 2 mm long; linear-lanceolate, base rounded auriculate, glabrous or blemished pilose, midrib stout, margin slightly scabrous.

In the process of target detection model training, dataset production and image annotation are two very important steps. It is the foundation of the dataset and can be directly related to the reliability of the experiment, while the accuracy of the image annotation directly affects the training effect and the accuracy of the test.

The sample dataset of maize leaf disease images selected and used in this paper is mainly collected from the open-source website PlantVillage (https://tensorflow.google.cn/datasets/catalog/plant_village accessed on 13 June 2022) for three common maize leaf diseases and healthy maize leaf images. The total number of datasets is 4353, in which maize maculatus, maize rust, and maize gray spot are the three common diseases of maize leaves listed in this paper. The three diseases and healthy leaves are shown in Figure 1.

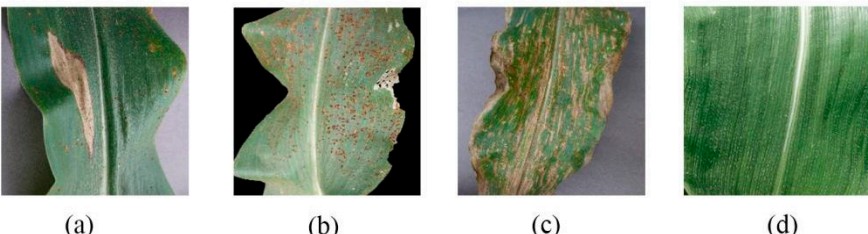

(a)                    (b)                    (c)                    (d)

**Figure 1.** Examples of maize leaf diseases. (**a**) Maize blotch disease; (**b**) Maize rust; (**c**) Maize gray spot disease; (**d**) Healthy maize leaves.

The number of maize leaf data sets is shown in Table 1.

Because of the small number of large spots and gray spots, this paper additionally takes the data set of maize leaves taken from the pear test field and expands the number of data sets of gray spot and large spots by cutting out 200 sheets of maize leaves with gray spot and 200 sheets of maize leaves with the large spot from the data set taken from the pear test field and filling them into the original data set.

**Table 1.** Number of maize leaf datasets.

| Leaf Type | Number of Images |
|---|---|
| Maize blotch disease | 1000 |
| Maize rust | 1191 |
| Maize gray spot disease | 1000 |
| Healthy maize leaves | 1162 |

The image annotation tool used in this paper is Make Sense (https://www.makesense.ai/ accessed on 13 June 2022), an online annotation tool recommended by the authors of YOLOv5n to annotate the images, which can directly output YOLO format label files and can be directly applied to the YOLOv5n network.

The sorted data set picture files and label files were divided into the training dataset, the validation dataset, and the test dataset according to the ratio of 6:2:2, and then put into the network model for subsequent data enhancement and model training.

### 2.2. Data Augmentation

When we want to obtain a well-performing neural network model, we must have a large amount of data to support it, but it takes a lot of time and labor to obtain new data. If we use data augmentation [17], we can use the computer to create new data to increase the number of training samples, for example, by changing color brightness, hue saturation, scaling, rotation, panning, cropping, perspective transformation, etc., and adding some appropriate noise data to improve the model generalization.

In the YOLOv5n network model described in this paper, not only are some basic data enhancement methods included, but also the Mosaic data enhancement [18] is used, whose main idea is to select four images from the used dataset, crop and scale them randomly, and then arrange them randomly to form a new image. This has the advantage of increasing the number of datasets while augmenting the number of small sample targets, and it improves the training speed of the model. The flowchart of Mosaic data enhancement is shown in Figure 2.

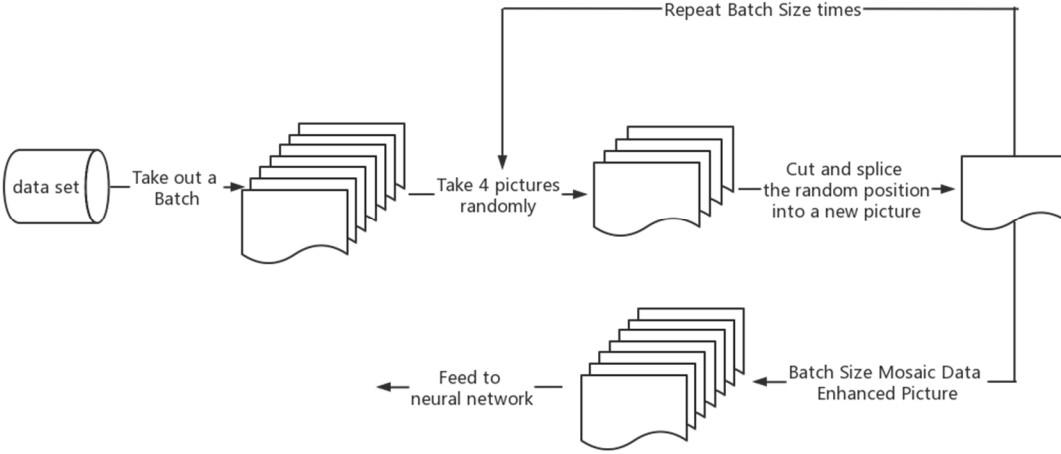

**Figure 2.** Mosaic data enhancement flow chart.

Mosaic data enhancement utilizes four images, which enriches the background of the detected objects and calculates the data of four images at once when BN calculates, so that the mini-batch size does not need to be large, and then a GPU can achieve better results.

In practice, Mosaic data enhancement first removes one batch of data from the total data set, takes out four images at random from it each time, crops and splices them at random positions, synthesizes new images, repeats the batch size several times, and finally gets a new batch size of one batch of images after mosaic data enhancement, then feeds to the neural network for training.

When cropping and splicing the images, the four randomly obtained images are cropped by a randomly positioned crosshair, and the corresponding parts are taken for splicing. At the same time, the target frame of each original image is limited by the crosshair crop, and will not exceed the original crop range. The implementation of Mosaic data enhancement in practice is shown in Figure 3.

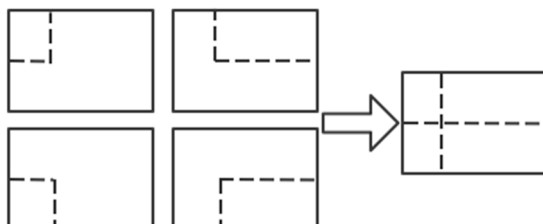

**Figure 3.** Implementation of Mosaic data enhancement in practice.

Mosaic has the following advantages: increases data diversity; randomly selects four images for combination; the number of images obtained from the combination is more than the number of original images; enhances model robustness; mixes four images with different semantic information; allows the model to detect targets beyond the conventional context; and enhances the effect of batch normalization. When the model is set to BN operation, the training will increase the total number of samples (BatchSize) as much as possible, because the BN principle is to calculate the mean and variance of each feature layer; if the total number of samples is larger, then the mean and variance calculated by BN will be closer to the mean and variance of the whole dataset, and the better the effect. The Mosaic data enhancement algorithm is helpful to improve the performance of small target detection. The enhanced images are stitched together from four original images, so that each image has a higher probability of containing small targets.

The operation principle of Mosaic data enhancement is equivalent to passing in four images for learning at one time during the training process, increasing the number of single training samples and target diversity, improving network training convergence speed and detection accuracy, and reducing large samples to small samples randomly, increasing the number of small-scale targets. Since the target of this paper is maize leaf disease and the disease spot is a small-scale target, Mosaic data enhancement provides important help for this study. Figure 4 shows 16 examples of data enhanced by Mosaic data. The file name in the picture is the file name of the image data involved in data enhancement. In the example, the file name is only for demonstration and will not be integrated into the picture to affect the subsequent model training. The colored boxes in the figure are identification boxes, where 0 indicates gray spot, 1 indicates rust, 2 indicates healthy maize leaves, and 3 indicates large spot disease. As shown in Figure 4.

To verify that the Mosaic data enhancement is real and effective for the experimental effect, this paper compares the parameters of the YOLOv5n network model with and without Mosaic data enhancement. The effect is shown in Figure 5.

From the Figure 5, it can be seen that the accuracy of the model is significantly and substantially improved after adding Mosaic data augmentation, and the convergence of the model is significantly improved compared to that without Mosaic data augmentation. At the same time, it can be seen that due to the Early Stopping method in the YOLOv5n model, which can resist overfitting, the model terminates early after 252 iterations in the training curve without the Mosaic data augmentation, because the accuracy no longer improves.

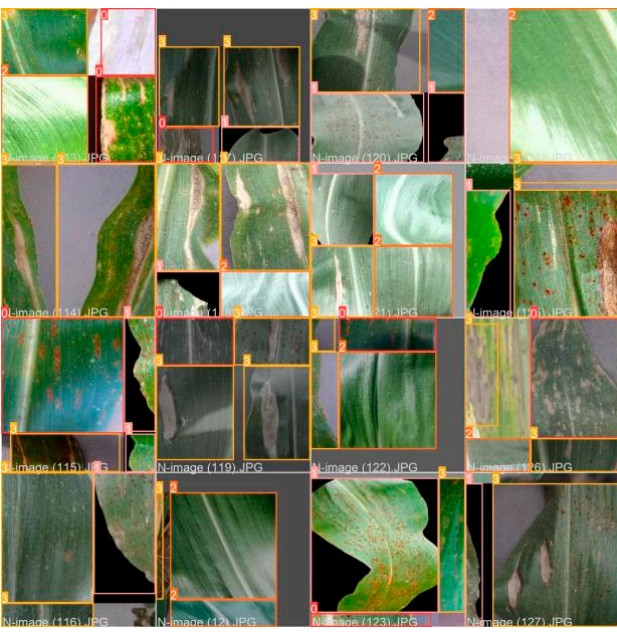

**Figure 4.** Sixteen examples of Mosaic data enhancement.

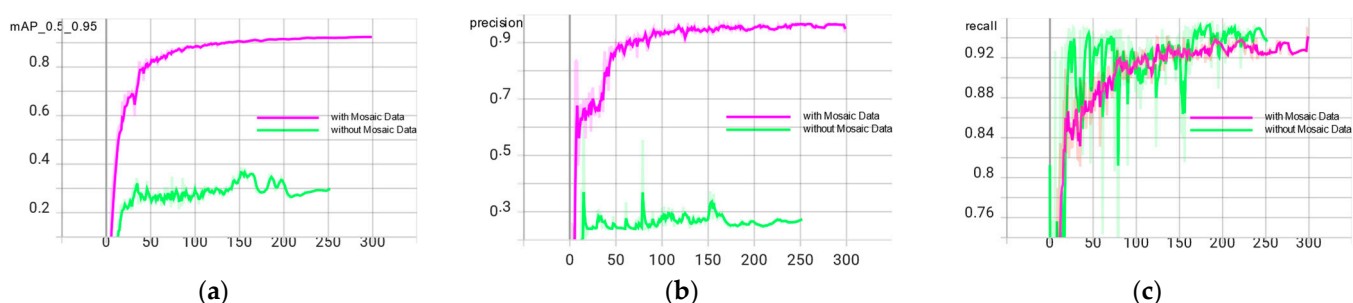

(**a**)  (**b**)  (**c**)

**Figure 5.** Comparison of enhancement effects with and without Mosaic data. (**a**) mAP_0.5_0.95 curve chart; (**b**) precision curve chart; (**c**) recall curve chart.

*2.3. YOLOv5 Network Model*

The YOLOv5 target detection algorithm is the 5th version of YOLO, whose core idea is to use the whole map as the input of the network and regress the location coordinates and category of the target directly in the output layer, which is characterized by high detection accuracy and fast detection speed to meet the demand of real-time monitoring.

The YOLOv5 network has been updated with five versions, YOLOv5n, YOLOv5s, YOLOv5m, YOLOv5l, and YOLOv5x in order, with similar network structures and changes in network depth and width of feature maps based on YOLOv5s. Its accuracy and inference speed follow, of which YOLOv5n is in 2021 October after YOLOv5 update version 6.0, which has the advantage of being the fastest and the smallest model size compared to other versions. The ultimate goal is to deploy the model to mobile for real-time detection. To meet the lightweight requirement, the final study of this paper decided to use the YOLOv5n detection model with the lowest complexity to reduce the model storage footprint and increase the recognition speed.

The YOLOv5n algorithm consists of four parts: input, backbone, neck, and prediction [19]. Among them, Mosaic data enhancement is beneficial for detecting small targets and is suitable for leaf disease identification in this paper. The adaptive image scaling operation fixes images of different sizes to 640 pixels × 640 pixels as input. In the backbone network, YOLOv5n mainly uses the Conv module CSP structure and SPPF module. The feature fusion stage mainly borrows the idea from PANet [20]. The FPN (Feature Pyramid Network) and PAN (Path Aggregation Network) are borrowed to form the FPN + PAN

structure. The prediction output continues the previous idea of YOLO by outputting three sizes of prediction maps at the same time, which are suitable for detecting small, medium and large targets. The network structure of YOLOv5n is shown in Figure 6.

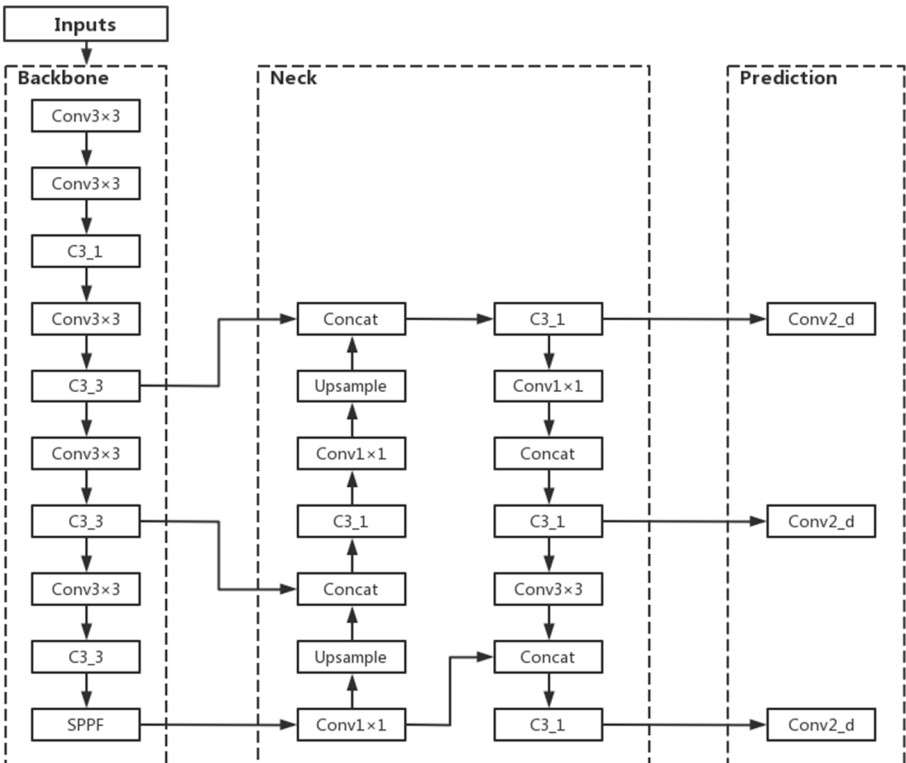

**Figure 6.** YOLOv5n network structure.

*2.4. Improvements to the YOLOv5n Model*

2.4.1. Adding CA to Improve Model Accuracy

In the task of maize leaf disease detection, since the disease spots occupy relatively few pixels of the image, their feature information is easily lost in the deep network, resulting in errors such as the wrong detection and missed detection. At this point, it would be more beneficial for the network model to recognize the images if the unsupervised network can automatically acquire the ability to focus on smaller pixel blocks. Therefore, this paper introduces the CA (Coordinate Attention) mechanism [21] in the YOLOv5n backbone network, which is used to tell the model "what" and "where," and which has been widely studied and deployed to improve the performance of neural networks. The use of lightweight attention modules can improve the network's ability to extract features from maize leaf spots while saving parameters.

For other channel attentions, they are taken to transform the input into individual feature vectors by 2D global pooling. The general idea of Coordinate Attention used in this paper is to decompose channel attention into two 1D feature encodings of aggregated features along different directions in the H-direction as well as the W-direction, that is, into $C \times H \times 1$ and $C \times 1 \times W$. CA This idea has the advantage of capturing long-range dependencies along one spatial direction while retaining accurate location information along the other spatial direction. After that, the generated feature maps are encoded separately, resulting in two direction-aware, as well as position-sensitive, feature maps, which can be complementarily applied to the input feature maps to enhance the representation of the target of interest. The two directional feature maps are then Concept spliced and then fed into a shared convolution to reduce the dimensionality to $C/r$, after which they are separated and allowed to Sigmoid in different directions to obtain the coefficients and then multiplied together. Finally, the feature map is obtained.

After adding CA attention to the YOLOv5n backbone network [22], keeping the parameters unchanged, the model is trained again, and the trained model has significantly improved the effect compared with the original model; the average accuracy mean value is increased from 0.924 to 0.948, and the model size is not significantly increased, which meets the requirement of being lightweight.

In this paper, after adding CA attention to the YOLOv5n backbone network, the specific structure of the backbone network is shown in Figure 7.

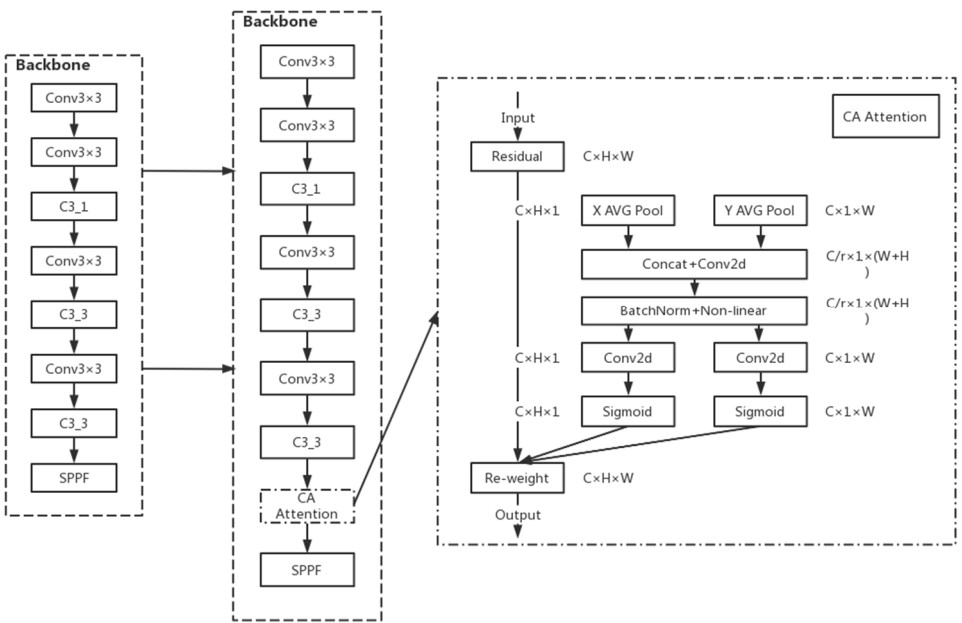

**Figure 7.** YOLOv5n's integration into CA attention mechanism.

2.4.2. Incorporating Swin Transformer Structure to Improve Model Generalization Performance

In the detection of maize leaf spots, the distribution of different types of spots in the leaf images differs: large spots occupy a small area of the leaf and rely more on local information of high-level features; rusts have a large distribution area and rely on global information more obviously; gray spots are moderate in size and rely on both local and global information. The performance of Convolutional Neural Networks (CNN) is more capable of capturing local information and has a certain disadvantage in global information acquisition. To alleviate the adverse effects of the non-uniform spot size, the model is improved by extracting global information using Swin Transformer [23]. In this paper, a smaller size target detection head was added to the original small, medium and large size detection heads of the YOLOv5n model to enhance its ability to identify small targets of the spots. The x-small size detection head in Figure 8, and the Swin Transformer structure, was incorporated into the large size detection head to replace the original C3 structure with the C3STR structure incorporated into the large size detection head to change the original C3 structure to C3STR structure, thus improving the model's capture of feature information.The improved network structure is shown in Figure 8.

The Swin Transformer model was proposed by Microsoft Research in 2021. Swin Transformer uses hierarchical feature maps similar to those used in convolutional neural networks, such as feature map sizes with 4×, 8×, and 16× down-sampling of images, such that the backbone helps to build on top of this for tasks such as target detection, instance segmentation, etc. The Swin Transformer network is another collision of the Transformer model in the field of vision. The Swin Transformer network is another collision of Transformer model in vision field.

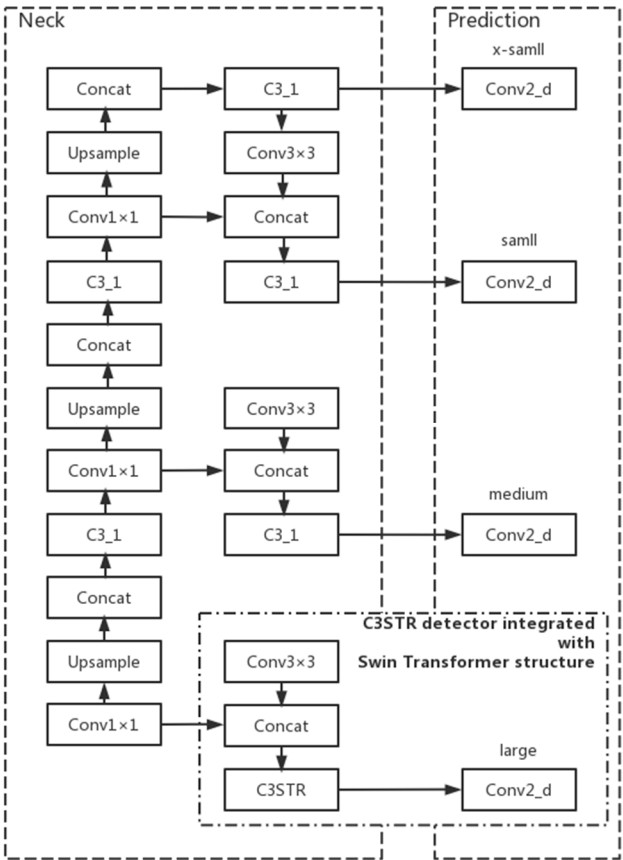

**Figure 8.** Add x-small detection head and Swin Transformer structure.

The concept of Windows Multi-Head Self-Attention (W-MSA) is used in Swin Transformer, for example, in the 4-fold downsampling and 8-fold downsampling in the figure below. The feature map is divided into multiple disjointed regions (Window), and Multi-Head Self-Attention is performed only within each window (Window).

The basic flow of the whole framework is as follows.

First, the image is input to the patch partition module for chunking, i.e., every $4 \times 4$ adjacent pixels is a patch, and then it is flattened in the channel direction. Assuming that the input is a three-channel RGB image, each patch has $4 \times 4 = 16$ pixels, and each pixel has three values, R, G, and B. The flattened image shape is $16 \times 3 = 48$, so the image shape changes from [H, W, 3] to [H/4, W/4, 48] after patch partition. Then the channel data of each pixel is linearly transformed by the linear embedding layer from 48 to C, i.e., the image shape is changed from [H/4, W/4, 48] to [H/4, W/4, C].

The Swin Transformer divides patches by first determining the size of each patch and then calculating the number of patches. The number of Swin Transformers decreases, and the perceptual range of each patch expands as the network depth deepens, which is designed to facilitate the construction of layers of Swin Transformer and to adapt to the multi-scale of visual tasks.The architecture of a Swin transformer (Swin-T) is shown in Figure 9.

Then the feature maps of different sizes are constructed by four stages; except for Stage 1, where a linear embedding layer is first passed, the remaining three stages are first downsampled by a patch merging layer. Note that there are two types of blocks, as shown in Figure 10, which differ only in that one uses the W-MSA structure and the other uses the SW-MSA structure. Moreover, these two structures are used in pairs, with one W-MSA structure used first and then one SW-MSA structure used.

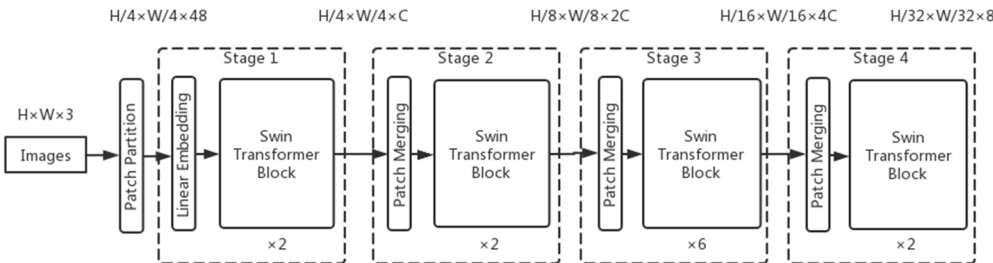

**Figure 9.** The architecture of a Swin Transformer (Swin-T).

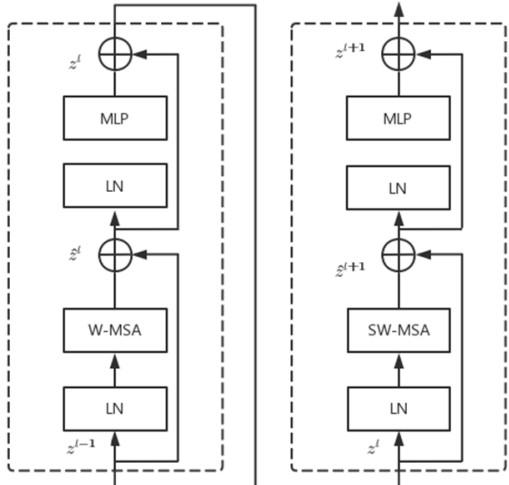

**Figure 10.** Two successive Swin Transformer blocks.

*2.5. Test Environment*

In this paper, we use the deep learning framework PyTorch to build and improve the model in the Anaconda3 environment, and train and test the model under the Windows 10 system. The computer CPU is 11th Gen Intel (R) Core (TM) i7–11700K @ 3.60 GHz and the GPU is NVIDIA GeForce GTX 1080 Ti. The GPU is used for acceleration to improve the network training speed, the Cuda version is 11.1.0, and the cudnn version is 8.1.0.

*2.6. Evaluation Metrics*

In this paper, the performance of the YOLOv5n network model is evaluated using several metrics from the target detection algorithm, specifically Precision (P), Recall (R), and mean Average Precision (mAP) [24]. Average Precision (AP) is the integral of the PR curve formed by taking Precision (P) as the vertical axis and Recall (R) as the horizontal axis. A recall is a metric that reflects the ability of the model to find positive sample targets, precision is used to reflect the ability of the model to classify samples, and average precision is a metric that reflects the overall performance of the model to detect and classify targets. The mean Average Precision (mAP) represents the average of the mean accuracy of all categories. Among all the metrics, mAP is the most important evaluation metric in the target detection algorithm, which can measure the accuracy of the detection algorithm. $mAP_{0.5}$ is the AP of the target detection model evaluated at an IoU threshold of 0.5. $mAP_{0.5}$ is its mean value for all categories; $AP_{0.5-0.95}$ is the mean value of the AP of the model evaluated at different IoU thresholds (0.5–0.95, step size 0.05); and $AP_{0.5-0.95}$ is the average value of AP evaluated under different IoU thresholds (0.5–0.95, step size 0.05), which is a more stringent model accuracy index. $mAP_{0.5-0.95}$ is the average value of all categories. In this paper, we choose $mAP_{0.5-0.95}$ as the evaluation index. The additional evaluation index considered in this paper is the number of parameters, and the number of parameters indicates the size of the storage space occupied by the model file in MB.

The expressions for the calculation of Precision (P), Recall (R), Average Precision (AP), and mean Average Precision (mAP) are shown in Equations (1)–(4).

$$P = \frac{TP}{TP + FP} \tag{1}$$

$$R = \frac{TP}{TP + FN} \tag{2}$$

$$AP = \int_0^1 P \cdot R dR \tag{3}$$

$$mAP = \frac{\sum_{i=1}^{N} AP_i}{N} \tag{4}$$

where:

-      The number of true samples.
-      The number of false positive samples.
-      Number of pseudo-negative samples.
-      The number of species in the sample.

The positive and negative samples are judged by setting the average Intersection over Union (IoU) threshold between the predicted target area and the actual target area, and if the IoU of both exceeds the threshold, the sample is positive, and if vice versa, the sample is negative.

## 3. Results

### 3.1. Model Trainsing Hyperparameter Setting

In the model training stage, the total number of training rounds is set to 300 and the iteration batch_size is set to 16. Setting too small a value for iteration batch size will lead to too slow training, and setting too large a value for iteration batch size will lead to insufficient video memory to run the experiment. In the experimental environment of this paper, the maximum YOLOv5x model runs at 83%, so setting a larger value will lead to run failure, so it is considered that the size of 16 is a more appropriate level.

In the model training process, if the learning rate is adjusted too large, the network will not converge, while if the learning rate is adjusted too small, the network convergence speed will become slow, so the appropriate learning rate is a key factor in the training process. This paper uses three different sizes of initial learning rate to compare the model experiments; the trained model parameters are shown in Table 2.

**Table 2.** Parameter comparison under different initial learning rates.

| Model | Initial Learning Rate | Precision (P) | Recall (R) | Mean Average Precision (mAP$_{0.5-0.95}$) | Parameter Quantity/MB |
|---|---|---|---|---|---|
| YOLOv5n | 0.1 | 0.938 | 0.924 | 0.885 | 3.9 |
| | 0.01 | 0.94 | 0.949 | 0.924 | 3.9 |
| | 0.001 | 0.963 | 0.922 | 0.889 | 3.9 |

From the table, it can be seen that, for the YOLOv5n model studied in this paper, the average precision mean is highest at an initial learning rate of 0.01. In addition, in order to see the comparison of network convergence more intuitively, this paper shows the graphs of model recall for three different sizes of initial learning rates, and the effect is shown in Figure 11.

From the figure, it can be seen that the model converges fastest when the learning rate is 0.01, the model converges best and performs more stably, and its accuracy stays around the optimal value after 100 epochs, which avoids the interference of the model fluctuation by chance on the final result; when the learning rate is 0.1, the model recall

curve fluctuates more and converges extremely poorly; when the learning rate is 0.001. Although the convergence effect is better when the learning rate is 0.001, the convergence speed is not as fast as when the learning rate is set to 0.01, and the accuracy remains around the optimal value after 150 epochs. Therefore, the initial learning rate of 0.01 is used to carry out the subsequent experiments.

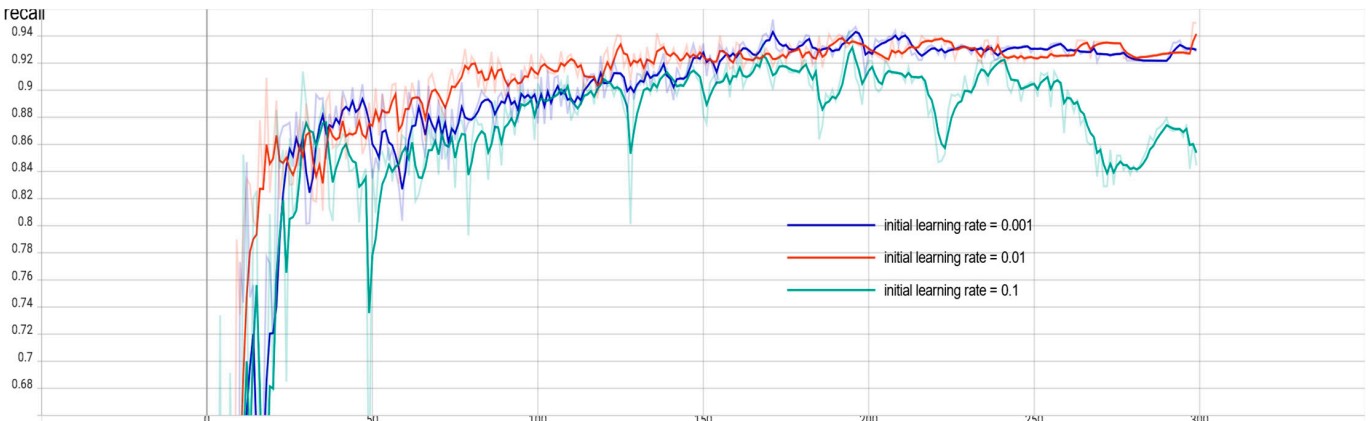

**Figure 11.** Comparison of recall rates under different initial learning rates.

To prevent the overfitting phenomenon, the weight decay coefficient is set to 0.0005, the confidence level is set to 0.5, and the non-maximum suppression threshold is set to 0.3.

### 3.2. Comparison of Different Algorithm Models

Since there are five versions of YOLOv5 so far, to ensure the authenticity, accuracy, and rigor of the experimental process, the five versions of YOLOv5n, YOLOv5s, YOLOv5m, YOLOv5l, and YOLOv5x are trained on the same platform, the same framework with the same training parameters, and on the same data set in turn. The comparative data are shown in Table 3.

**Table 3.** Comparison of parameters of different network models.

| Model | Precision (P) | Recall (R) | Mean Average Precision (mAP$_{0.5-0.95}$) | Parameter Quantity/MB |
|---|---|---|---|---|
| YOLOv5n | 0.94 | 0.949 | 0.924 | 3.9 |
| YOLOv5s | 0.973 | 0.942 | 0.932 | 14.5 |
| YOLOv5m | 0954 | 0.961 | 0.945 | 40.3 |
| YOLOv5l | 0.96 | 0.965 | 0.948 | 92.9 |
| YOLOv5x | 0.965 | 0.965 | 0.958 | 173.2 |

From the above table, we can see that, due to the different depth and width of the models, among the five models of YOLOv5, the average accuracy of YOLOv5n is 0.924, which is the lowest accuracy among the five models, but its model occupies much less memory than the other models, only 3.9 M, and the training time of YOLOv5n is also much shorter than the other models during training. In contrast, the memory consumption of YOLOv5l and YOLOv5x models is around 100 MB after training with smaller data sets, which can be considered as suitable for deployment on large servers only.

Also, among the five models of YOLOv5, YOLOv5n has the lowest precision, but its model takes up far less memory than the other models, only 3.9 M, and the training time of YOLOv5n is also much shorter than the other models. In contrast, the memory consumption of the YOLOv5l and YOLOv5x models is around 100 MB after training with a small data set, which can be considered suitable for deployment on large servers only.

The ultimate goal of this paper is to help farmers accurately identify the diseases infecting maize leaves in real-time by using portable cell phones in the field so that timely

control can be carried out to reduce the adverse effects of low quality and yield of maize caused by the diseases.

Therefore, YOLOv5n is the most suitable lightweight model for mobile deployment to identify maize leaf spots in real-time. The model is also improved in the hope that the accuracy of the YOLOv5n model can be further enhanced to reach a similar level to other large network models without affecting the model size.

### 3.3. Model Enhancement by Different Attention Mechanisms

In this paper, the CTR_YOLOv5n model is built by incorporating CA attention into the baseline model and using the Swin Transformer-based C3STR block as the prediction head to further improve the generalization ability of the model and improve the model accuracy.

To verify the effectiveness of the improvements on the model, this paper adds the current mainstream attention mechanisms, such as SE (Squeeze-and-Excitation), CBAM (Convolutional Block Attention Module), and ECA (Efficient Channel Attention), to the model under the same platform and framework, using the same training parameters, and the comparison results are shown in Table 4.

**Table 4.** Comparison of Yolov5n effect improvement under different attention mechanisms.

| Model | Precision (P) | Recall (R) | Mean Average Precision (mAP$_{0.5-0.95}$) | Parameter Quantity/MB |
|---|---|---|---|---|
| YOLOv5n | 0.94 | 0.949 | 0.924 | 3.9 |
| SE_YOLOv5n | 0.946 | 0.959 | 0.931 | 3.9 |
| CBAM_YOLOv5n | 0.962 | 0.947 | 0.927 | 3.9 |
| ECA_YOLOv5n | 0.961 | 0.944 | 0.931 | 3.9 |
| CA_YOLOv5n | 0.948 | 0.961 | 0.948 | 3.9 |

From the above table, we can see that the addition of the attention mechanism to the YOLOv5n backbone network improves the mean accuracy of YOLOv5n, and there is no significant change in the model size, which proves that the attention mechanism has an improvement effect on YOLOv5n. Among the mainstream attention mechanisms, the SE attention mechanism improves the mean accuracy of YOLOv5n by 0.7 percentage points, CBAM by 0.3 percentage points, ECA by 0.7 percentage points, and the improved CA_YOLOv5n by 2.4 percentage points. In this study, the accuracy of the CA_YOLOv5n network model is improved to the same level as YOLOv5l while keeping the model size the same as YOLOv5n, confirming that the improvements in this paper can allow CA_YOLOv5n to be deployed on mobile devices with an accuracy similar to that of the server-deployed model.

In order to see the improvement effect of different attention mechanisms on YOLOv5n more intuitively, this paper shows the accuracy comparison of YOLOv5n with different attention mechanisms added, and the effect is shown in Figure 12.

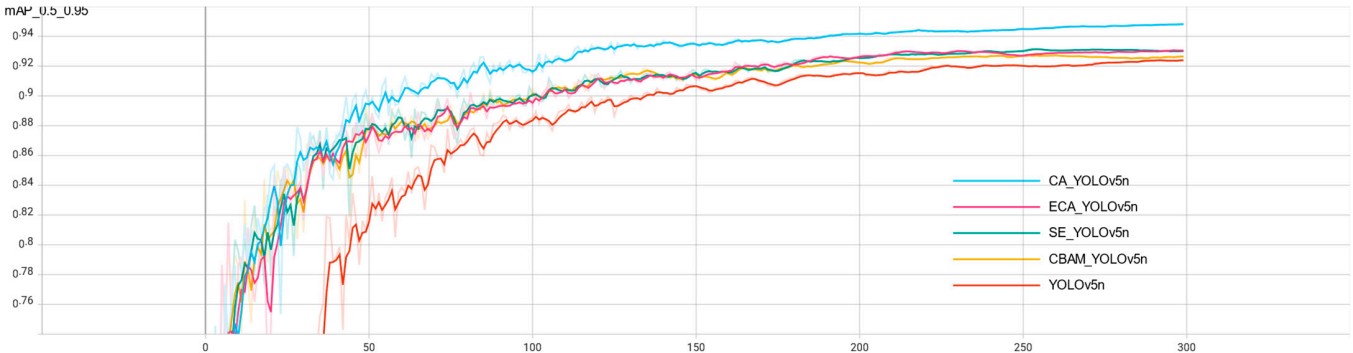

**Figure 12.** Precision comparison of different attention mechanisms.

From the figure, we can see that the accuracy of the YOLOv5n network without the attention mechanism starts to increase rapidly around 40 training sessions, and the increase in accuracy starts to slow down around 100 training sessions, while the accuracy of the model with the attention mechanism starts to increase rapidly around 15 training sessions, in which CA_YOLOv5n has the fastest growth rate and the largest increase in the first 50 training sessions, and the accuracy is always higher than the other models in the subsequent training sessions. The accuracy of CA_YOLOv5n is consistently higher than the other models in the subsequent training.

Comparing CA with other mainstream attention mechanisms, CA attention takes into account not only the relationship between channels but also the location information in the feature space. CA uses a more efficient method to capture location information and channel relationships. It does this by decomposing the two-dimensional global pooling operation into two directions, width and height, and then averaging the pools globally to obtain feature maps in the two different directions, width and height. The feature maps in both directions are then processed to obtain the final feature maps, which is the main reason why the CA attention is more effective than other attentions.

The experiments in this paper can prove that the model improvement approach studied in this paper is better than adding other mainstream attention mechanisms. In addition, from Figure 6, we can see that all the models reach a relatively flat accuracy after 200 training sessions, which indicates that the proposed model achieves the ideal state.

### 3.4. Model Improvement to Improve the Effect

Based on the CA_YOLOv5n model, the experiments in this paper make further improvements by adding a small target detection layer to the neck network neck of the CA_YOLOv5n model to enhance its ability to recognize small targets of disease spots, and by replacing the original C3 detection head into the Swin Transformer structure with the C3STR detection head to improve the model's capture of feature information.

The following Table 5 shows the integrated inference results of each model for different types of maize leaf diseases at $256 \times 256$ pixels. The YOLOv5n model showed high recognition ability for maize rust and healthy maize leaves, reaching 0.953 and 0.99, respectively, but low recognition rate for maize gray spot disease and maize blotch disease, reaching only 0.905 and 0.849. The overall accuracy of the model was further improved by adding the CA attention mechanism. Compared with YOLOv5n, the model CA_YOLOv5n showed a significant improvement in the recognition performance of maize gray spot disease and maize blotch disease, which was comparable to the performance of YOLOv5l model. The final proposed model, CTR_YOLOv5n with $mAP_{0.5–0.95}$, improved by 2.8% compared to the initial model YOLOv5n and by 0.4% compared to the YOLOv5l model. It is noteworthy that the $mAP_{0.5–0.95}$ of model CTR_YOLOv5n reached 0.928 and 0.925 for maize gray spot disease and maize blotch disease, which were significantly improved compared with each model, proving that the accuracy of the final model proposed in this paper is effectively improved for small target detection of disease spots. The accuracy distribution of the CTR_YOLOv5n model is more balanced, which shows the effectiveness of the improvement strategy of this paper. Although the number of parameters in the final model is slightly increased compared with YOLOv5n, it still meets the requirements of being lightweight.

**Table 5.** Performance of each model under different disease categories.

| Model | Maize Gray Spot Disease | Maize Rust | Healthy Maize Leaves | Maize Blotch Disease | Mean Average Precision ($mAP_{0.5–0.95}$) | Parameter Quantity/MB |
|---|---|---|---|---|---|---|
| YOLOv5n | 0.905 | 0.953 | 0.99 | 0.849 | 0.924 | 3.9 |
| YOLOv5l | 0.916 | 0.986 | 0.992 | 0.898 | 0.948 | 92.9 |
| CA_YOLOv5n | 0.924 | 0.985 | 0.992 | 0.891 | 0.948 | 3.9 |
| CTR_YOLOv5n | 0.928 | 0.964 | 0.99 | 0.925 | 0.952 | 5.1 |

It is confirmed that the improvements in this paper allow CTR_YOLOv5n to be deployed on mobile with an accuracy comparable to that of the server-deployed model.

## 4. Discussion

Crop disease identification has an important position in the field of digital agricultural information technology as well as in the construction of smart agriculture. In addition, the existing crop disease recognition has many kinds, low accuracy, and poor targeting, which cannot meet the actual needs of maize growers. Deep learning, as an emerging technology in the field of machine learning, has a wide range of applications in the field of image recognition.

In this paper, we study the improved CTR_YOLOv5n model to identify and detect maize leaf diseases, identify maize blotch disease, gray spot, and rust, provide technical reference for maize disease control, and help farmers to solve the problem of difficult maize disease identification, which has high application value. The main findings are as follows.

1. Since disease spots generally occupy a small leaf area, as a small-scale target, the number of data sets may not be sufficient. Using the mosaic data enhancement in the YOLOv5n model, the large samples were randomly reduced to small samples, increasing the number of small-scale targets. The number of datasets was effectively increased to speed up the network convergence and improve the model accuracy.

2. To address the characteristics that it is not easy to distinguish between different diseases of maize leaves, especially those with similar color and shape between maize blotch disease and maize gray spot disease, the backbone network of the YOLOv5n model is improved to incorporate the CA attention mechanism to enhance the extraction of spot features, add a minimum size detection head of YOLOv5n model, and introduce a Swin Transformer structure in the large size detection head to fuse global and local information to improve the generalization performance of the model. This improves the generalization performance of the model and, finally, improves the recognition accuracy by 2.8%.

3. Using the CTR_YOLOv5n model to identify maize leaf spots, the average recognition accuracy is 95.2%, which is 2.8% higher than the original model; compared with the YOLOv5l model, the trained model size is maintained at 5.1MB with the same accuracy, which is 94.5% smaller. The model accuracy is effectively improved with a small model footprint. It can be concluded that the improved CTR_YOLOv5n model is more suitable for real-time detection of maize leaf diseases on mobile applications and improves detection efficiency.

The results show that the improved CTR_YOLOv5n model helps to improve the overall effect of maize leaf disease recognition. Considering that the image background is often complex in actual disease recognition, whether the model can still guarantee such a high accuracy rate for complex backgrounds and occluded cases is subject to further research. In addition, the scope of this paper is limited to the identification of diseases on the leaves of maize. In addition, the scope of identification can be extended to the identification of diseases on the roots and ears of maize, such as maize stalk rot, maize silky black ears, and maize rot, which have a high incidence rate and can cause huge losses, and can threaten the life and health of humans and animals in serious cases.

## 5. Conclusions

In this paper, we propose a method to improve the YOLOv5n model for the recognition and detection of maize leaf spots. CA attention is incorporated into the backbone network of the YOLOv5n model to increase the weight of feature information of maize leaf spots, so that the model pays more attention to the feature information of spots during training; a smaller size target detection head is added to enhance its ability to recognize small targets of spots; and the Swin Transformer structure is incorporated into the large size detection head, so as to improve the model's ability to capture feature information. The accuracy of the model for maize leaf spot recognition is effectively improved. The experimental results show that the average recognition accuracy of the proposed method can reach 95.2%

for maize leaf spots, and the recognition accuracy of maize gray spot and maize large spot disease, which are not easy to recognize, has been effectively improved, reaching 92.8% and 92.5%, respectively. Compared with other deep learning network models, this method has the advantages of higher accuracy and smaller size. In the future, we will collect some images of maize leaf diseases in natural environments and improve the model in a more lightweight way to develop a crop disease recognition system that can be applied to mobile devices.

**Author Contributions:** The contributors are J.Z. and Q.Y. for conceptualization; H.Y. for methodology; J.Z. for formal analysis; L.M. and Q.Y. for investigation/writing—original draft/supervision; L.M. and J.Z. for visualization; J.Z. for writing—review/editing. All authors have read and agreed to the published version of the manuscript.

**Funding:** This work was supported by the National Natural Science Foundation of China–Joint Fund (u19a2061), Jilin Provincial Department of Education Project (No. JJKH20210336KJ), Jilin Province Ecological Environment Department Project (2021–07), Jilin Province Science and Technology Development Plan Project (No. 20200301047RQ), and Jilin Provincial Natural Science Foundation (No. 20200201288JC).

**Data Availability Statement:** Data supporting the findings of this study are available from the corresponding author.

**Conflicts of Interest:** The authors declare no conflict of interest.

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
