# Peer review of "Maize Leaf Disease Identification Based on YOLOv5n Algorithm Incorporating Attention Mechanism"

_agronomy, doi:10.3390/agronomy13020521_

Round 1

Reviewer 1 Report

The authors presented a very interesting manuscript whose method is very interesting. However, the authors did themselves by mixing the methods, results and discussion sections. Most of what is called the results is actually the materials and methods, and the discussion is the actual results. This essentially there is barely any discussion to talk about. May the authors work on these inconsistencies.

Author Response

请参阅附件。

Reviewer 2 Report

The authors Ma et al. presented a manuscript on enhancing the YOLOv5n algorithm with an attention mechanism for leaf disease detection in the maize crop. Overall, I like the topic of the implementation of deep learning into corn leaf disease management from a mobile app, and I do recognize that since it’s developed for mobile applications, the lightweight and high accuracy of the model is critical for such work. However, I have several major concerns regarding this manuscript and I suggest the authors extensively revise the manuscript before it’s considered for publication.

(1) I understand the major focus of the study is due to the many samples to test with many different disease types, there is a higher chance of inaccuracy in the identification process. However, after reading the manuscript, I do not see clearly what is your innovation and contribution to addressing this major challenge. The data is open source, the training set is open source, Mosaic Data Enhancement is also open source, and YOLOv5n is also open source. YOLOv5n to CA attention is also an existing method. See my comment 2.

(2) Several papers are already implementing the YOLOv5n to CA attention.

https://pubag.nal.usda.gov/catalog/7931025

https://www.sciencedirect.com/science/article/pii/S0168169922006718

(3) The authors need a thorough explanation of how mosaic data enhancement works. The substantial description that the authors put about the method is literally two sentences “crop and scale them randomly, and then arrange them randomly to form a new image. This has the advantage of increasing the number of datasets while augmenting the number of small sample targets, and it improves the training speed of the model.” I am not convinced, after reading this, that mosaic data enhancement will work that well. As the authors reported in figure 4a and figure 4b, the enhancement is even 4 times between with/without mosaic. How could cropping and scaling on the existing image enhance the training by this great?

Specific comments:

Line 52 to 63. The literature work on state-of-art methodology is weak. Please expand your literature review to cover the maize crop. Ginger and Melon are not great references for your research topic unless you could not find any reference on maize.

Line 242 please define FPN and PAN for the first time use of the term.

Line 396 to 399 please fill “--" with your variables

Check the reference style [] throughout the paper. It looks like you used superscription.

Line 457 to 484: the inclusion of YOLO V4 diminishes the focus of the paper. Since YOLOv5n is your target, probably delete this section unless you can provide a strong justification.

Line 499 to 545: It won’t surprise the reader if you just mention that YOLOV5n +CA works better. As correspond with my point, since this is not the first time using YOLOV5n +CA, I would suggest you discuss the improvement you get with your case as compared with other research using YOLOV5n +CA, even though they are for different crops. 

Reviewer 3 Report

In this study, a YOLO-based classification system is proposed to identify common leaf spot, gray spot and rust diseases in maize. It has been observed that the proposed structure contains innovations. However, it should be arranged taking into account the following points.

1-      Is the image presented in figure 3 a single augmented record or a presentation of 16 augmented record? Are there filenames on your records as seen in figure 3? If there are no file names on the photos, this should be clearly stated in the explanation of figure 3. If images have text on them, how do you make sure that these characters don't bias machine learning in disease classification?

2-      More related state-of-the-art references working on leaf classification studies (like; Azlah, M. A. F., Chua, L. S., Rahmad, F. R., Abdullah, F. I., & Wan Alwi, S. R. (2019). Review on techniques for plant leaf classification and recognition. Computers, 8(4), 77.; Koklu, M., Unlersen, M. F., Ozkan, I. A., Aslan, M. F., & Sabanci, K. (2022). A CNN-SVM study based on selected deep features for grapevine leaves classification. Measurement, 188, 110425.; Argüeso, D., Picon, A., Irusta, U., Medela, A., San-Emeterio, M. G., Bereciartua, A., & Alvarez-Gila, A. (2020). Few-Shot Learning approach for plant disease classification using images taken in the field. Computers and Electronics in Agriculture, 175, 105542. etc.) recommended to be surveyed and compared in this paper.

3-      By which ratios was the dataset divided as train dataset, validation dataset and test dataset?

4-      Was data augmentation performed before or after the database was split into training, validation, and testing?

Round 2

Reviewer 2 Report

After careful review, the authors addressed my concern and the manuscript is in good shape.  I agree with acceptance in its current form.

Reviewer 3 Report

Appropriate responses were given to all criticisms. I think it is more understandable and interesting for readers with the final version. The article can be published as it is.